# Synthesising Counterfactual Explanations via Label-Conditional Gaussian Mixture Variational Autoencoders

**Junqi Jiang**[1,2*]**, Francesco Leofante**[1]**, Antonio Rago**[1,3]**, Francesca Toni**[1]
[1]Imperial College London    [2]J.P. Morgan AI Research    [3]King's College London
{junqi.jiang, f.leofante, a.rago, f.toni}@imperial.ac.uk

## Abstract

Counterfactual explanations (CEs) provide recourse recommendations for individuals affected by algorithmic decisions. A key challenge is generating CEs that are *robust* against various perturbation types (e.g. input and model perturbations) while simultaneously satisfying other desirable properties. These include *plausibility*, ensuring CEs reside on the data manifold, and *diversity*, providing multiple distinct recourse options for single inputs. Existing methods, however, mostly struggle to address these multifaceted requirements in a unified, model-agnostic manner. We address these limitations by proposing a novel generative framework. First, we introduce the Label-conditional Gaussian Mixture Variational Autoencoder (**L-GMVAE**), a model trained to learn a structured latent space where each class label is represented by a set of Gaussian components with diverse, prototypical centroids. Building on this, we present **LAPACE** (LAtent PAth Counterfactual Explanations), a model-agnostic algorithm that synthesises entire paths of CE points by interpolating from inputs' latent representations to those learned latent centroids. This approach inherently ensures robustness to input changes, as all paths for a given target class converge to the same fixed centroids. Furthermore, the generated paths provide a spectrum of recourse options, allowing users to navigate the trade-off between proximity and plausibility while also encouraging robustness against model changes. In addition, user-specified *actionability* constraints can also be easily incorporated via lightweight gradient optimisation through the L-GMVAE's decoder. Comprehensive experiments show that LAPACE is computationally efficient and achieves competitive performance across eight quantitative metrics.

## 1 Introduction

Counterfactual Explanations (CEs) are a prominent method in explainable AI, illustrating the minimal changes needed for an input to achieve a different, more desirable prediction from a machine learning model (Wachter et al., 2017; Tolomei et al., 2017; Dwivedi et al., 2023). CEs are especially useful for providing algorithmic recourse, such as offering actionable suggestions to a bank customer who was denied a loan by an algorithmic decision-making system. An ideal CE should satisfy several properties: it must be *valid* (achieving the desired outcome), *proximal* (close to the original input), *plausible* (residing within the data manifold), and part of a *diverse* set of options (we refer the reader to Guidotti (2022); Karimi et al. (2023) for recent overviews).

In addition to these standard properties, various forms of *robustness* have been identified as a critical challenge (Upadhyay et al., 2021; Dominguez-Olmedo et al., 2022; Jiang et al., 2023; 2024a; Leofante & Wicker, 2025). Research in this area has been fragmented, often addressing specific robustness types in isolation or without considering their interplay with other key criteria (Jiang et al., 2024b). This has resulted in a lack of methods that can generate CEs integrating multiple properties, such as validity, plausibility, diversity, and multiple forms of robustness, within a single framework. This paper introduces a novel approach to bridge this gap, with a specific focus on (i) encourag-

---

*Work done while at Imperial College London.

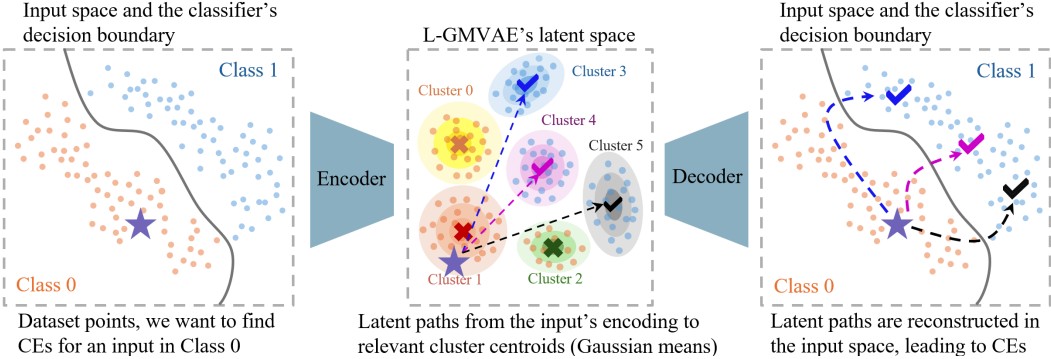

Figure 1: Illustration of LAPACE in binary classification. Given a dataset with a trained classifier's predictions (Left), a L-GMVAE is first learned with latent clusters (Gaussian components), capturing the data distribution with the classifier's predictions. In this example, we have 6 Gaussian components (Middle, the coloured areas). Prediction label 0 (1) is associated with Clusters 0-2 (3-5). The cluster centroids (learned Gaussian mixture prior) for classes 0 and 1 are marked with crosses and check marks. Assuming we are computing CEs for a negatively classified point (Left, purple star), LAPACE first performs linear interpolations linking the input's latent representation to each class 1 cluster centroid (Middle, dashed lines). These paths are then decoded to the input space to obtain paths of points, where they terminate at the decoded class 1 cluster centroids (Right).

ing robustness to model changes, such that the recourse stays valid under model parameter changes (such as regular retraining), and (ii) guaranteeing robustness to input perturbations, which amounts to the stability of generated CEs against small perturbations applied to the input (Artelt et al., 2021; Leofante & Potyka, 2024). Indeed, if a method finds CEs that are readily invalidated under some model updates, or provides significantly different CEs for nearly identical inputs, the CEs are likely based on arbitrary model behaviour and algorithmic artefacts, rather than meaningful features. This particularly undermines CEs' trustworthiness for recourse purposes.

A recent state-of-the-art method enhances robustness to input changes by leveraging diverse instances of nearest neighbours in the training dataset from the target class (Leofante & Potyka, 2024). However, this method remains heuristic and cannot guarantee perfect stability given the randomness in its distance-based thresholding process, and risks exposing sensitive training data. We argue that a more principled approach to robustness involves identifying a set of diverse, prototypical recourse points for the target class and then guiding all generated CEs to converge towards these points. While prior works have used generative models like standard variational autoencoders (VAEs), normalising flows, or diffusion models with similar intuitions, their primary goal was to improve plausibility by obtaining a realistic data manifold (Pawelczyk et al., 2020a; Wielopolski et al., 2024; Na & Lee, 2025). A key limitation is that they are typically unconditional, ignoring the information encoded in a classifier's predicted labels that are readily available during recourse generation. As a result, sampling with randomness, or complex algorithms such as latent space gradient optimisation, is needed to perform the CE search in the latent space, without guaranteeing their validity.

Based on these insights, we propose Label-conditional Gaussian Mixture VAE (**L-GMVAE**), a novel generative model for recourse. We adapt the standard GMVAE (Kingma et al., 2014; Dilokthanakul et al., 2016; Shu, 2016) to be label-conditional (Sohn et al., 2015) by explicitly mapping each class label to a dedicated set of Gaussian clusters in the latent space. After training, the mean of each Gaussian component serves as a latent centroid. As these centroids are learned through the VAE's objective, their decodings act as diverse, plausible, and robust prototypes for their associated class, making them ideal targets for recourse.

Building on this model, we introduce **LAPACE** (LAtent PAth Counterfactual Explanations), a simple yet effective method for generating paths describing smooth trajectories that connect inputs to their CEs. This method is illustrated in Figure 1. For any input, LAPACE finds a latent representation and generates multiple paths by linearly interpolating to each of the target class's latent centroids. When decoded, these latent trajectories form paths of high-quality CEs in the input space. Because all paths for a given target class converge to the same set of fixed prototypes, LAPACE achieves per-

fect robustness to input perturbations. Furthermore, providing paths of CEs allows users to choose between solutions that are close to their original input and those that are more robust but further away. In addition, LAPACE is model-agnostic, requiring only black-box access to the classifier's prediction function. The synthetic CEs also do not expose real dataset points.

Overall, we make the following contributions: **(1)** We propose L-GMVAE, a new generative model tailored for generating high-quality CEs (Section 3.1). **(2)** We introduce LAPACE, a novel model-agnostic CE generation method that uses the L-GMVAE to produce paths of CEs that are robust to input perturbations while also addressing validity, proximity, plausibility (our strong plausibility also leads to robustness against model changes), and diversity (Section 3.2). **(3)** We show how user-specified actionability constraints can be easily incorporated into LAPACE (Section 3.3). **(4)** We conduct comprehensive experiments that validate the competitive performance of LAPACE across eight quantitative metrics (Section 4).

## 2 RELATED WORK

**Counterfactual explanations** are typically computed using specialised algorithms with gradient descent (Wachter et al., 2017) or mixed integer programming (Mohammadi et al., 2021) to address validity and proximity. To enhance these explanations, plausibility is often enforced by aligning CEs with the data manifold through nearest-neighbour approaches (Brughmans et al., 2023; Poyiadzi et al., 2020) or by using generative models (Mahajan et al., 2019; Pawelczyk et al., 2020a; Wielopolski et al., 2024; Na & Lee, 2025). Diversity can be incorporated through multi-objective optimisation (Mothilal et al., 2020; Dandl et al., 2020) or creating variations in a certain sampling process (Leofante & Potyka, 2024). More recently, various forms of robustness have been identified as a critical property of CEs (Mishra et al., 2021; Jiang et al., 2024b). These include robustness to input changes, ensuring similar inputs receive similar explanations (Artelt et al., 2021), and robustness to model changes, which requires CEs to remain valid after model updates like retraining or parameter shifts (Upadhyay et al., 2021; Dutta et al., 2022; Jiang et al., 2024c). This is often encouraged by pushing the CE further away from the classifier's decision boundary into some more plausible regions (Pawelczyk et al., 2020b). We refer the reader to recent surveys (Guidotti, 2022; Karimi et al., 2023; Laugel et al., 2023; Jiang et al., 2024b) for in-depth reviews. Unlike most existing methods which address these properties in isolation, our proposed method is designed to simultaneously address a comprehensive set of these properties within a single, unified framework.

Parallel to the CE research in the explainable AI literature which concerns revealing faithfully what changes would be needed for *a given classifier* to flip its prediction, there have been substantial research efforts on causal inference, that instead aim at causally characterising the underlying *data generation process* (Pawlowski et al., 2020; Shen et al., 2022; Ribeiro et al., 2023; Monteiro et al., 2023; Wu et al., 2025; Ribeiro et al., 2025). A causal model is often explicitly considered, allowing counterfactual inference (Pearl, 2009) and thereby discovering causes behind certain observations (Halpern, 2016). From a cognitive science perspective, CEs have been shown to serve complementary goals for end-user perception compared to the causal explanations (Byrne, 2019), highlighting the inherent differences between the two fields. Our work follows the standard CE setups (causality-agnostic) introduced above, and incorporating causal constraints into the CE generation process (as studied by Karimi et al. (2020; 2021)) would be exciting future work.

**Learning latent space with clusters.** Early approaches, such as (Xie et al., 2016), learn low-dimensional embeddings for clustering using an autoencoder structure but lacked a generative component for sampling new data. VAEs (Kingma & Welling, 2014) provide a natural framework for this task by imposing a Gaussian prior on the latent space variables. The M2 model by Kingma et al. (2014) extends this idea to semi-supervised learning, formulating a mixture VAE where each latent component is explicitly tied to a class label. Building on this, the Gaussian Mixture VAE (GM-VAE) (Dilokthanakul et al., 2016) introduces a categorical latent variable to model cluster assignments, thus encouraging structured latent spaces. A simplified variant by Shu (2016) incorporates the assignment variable directly into the encoder, yielding more effective clustering. In parallel, Conditional VAEs (Sohn et al., 2015) generalise the framework by conditioning both inference and generation on auxiliary attributes. Our proposed L-GMVAE combines these perspectives: it learns a Gaussian mixture–structured latent space while conditioning on predicted label information.

## 3 LATENT PATH COUNTERFACTUAL EXPLANATIONS

In this section, we first present L-GMVAE, discussing how it is inherently suitable for synthesising CEs. Then, we present LAPACE, and show how actionability requirements can be accommodated via gradient updates through the decoder network.

**General notation .** For classification tasks, given an input $x \in \mathcal{X} \subseteq \mathbb{R}^d$ and a set of $L$ discrete labels $\mathcal{Y} = \{1, \ldots, L\}$, a classification model is a function that maps an input to a label, i.e., $M : \mathcal{X} \to \mathcal{Y}$. We denote the discrete prediction variable as $y \in \mathcal{Y}$, and model predictions as $M(x) = y$. Furthermore, we refer to a classification dataset with $N$ points as $\mathcal{D} = \{x^i, y^{*,i}\}_{i=1,\ldots,N}$, where each $y^* \in \mathcal{Y}$ is the ground-truth label. For an input $x$ and a desirable label $y' \in \mathcal{Y}$, $y' \neq M(x)$, a counterfactual explanation is $x' \in \mathcal{X}$ such that $M(x') = y'$, and $x'$ is close to $x$ as measured by some distance metric(s). L1 is commonly used to induce sparse changes (Wachter et al., 2017).

### 3.1 LABEL-CONDITIONAL GMVAE

**L-GMVAE.** Throughout the CE generation pipeline, the classifier-predicted label for the input and the desirable label for CEs are known. For computing CEs, it would therefore make sense to explicitly incorporate label information into the cluster assignments. Following GMVAE ((Shu, 2016)'s version, see Appendix A), the latent variable $z \in \mathbb{R}^h$ governs the data generation process. $z$ is regulated by a Gaussian Mixture with $K$ components, with a discrete variable $c \in \mathcal{C} = \{1, \ldots, K\}$ controlling the cluster assignment. In L-GMVAE, we partition the Gaussian clusters $\mathcal{C}$ using the possible $L$ labels in $\mathcal{Y}$ as: $\mathcal{C} = \mathcal{C}_1 \cup \mathcal{C}_2 \cup \ldots \cup \mathcal{C}_L$, where each $\mathcal{C}_i \subset \mathcal{C}$, and for all $\mathcal{C}_i, \mathcal{C}_j$ with $i \neq j$ and $i, j \in \mathcal{Y}$, $\mathcal{C}_i \cap \mathcal{C}_j = \emptyset$. The most intuitive approach for partitioning is to uniformly assign $K/L$ clusters for each class. Then, for a given class label $y$, its corresponding clusters are $\mathcal{C}_y$.

The generative component of L-GMVAE, $p(x, c, z \mid y)$, is characterised as:

$$p(x, c, z \mid y) = p(c \mid y)\, p_\theta(z \mid c)\, p_\theta(x \mid z) \tag{1a}$$

$$p(c \mid y) = 1/|\mathcal{C}_y| \text{ if } c \in \mathcal{C}_y \tag{1b}$$

$$p_\theta(z \mid c) = \mathcal{N}(z \mid \mu_z(c), \sigma_z^2(c)) \tag{1c}$$

$$p_\theta(x \mid z) = \mathcal{N}(x \mid \mu_x(z), \sigma_x^2(z)) \tag{1d}$$

The inference model $q(z, c \mid x, y)$ of the GMVAE is factorised as:

$$q(z, c \mid x, y) = q_\phi(c \mid x, y)\, q_\phi(z \mid x, c, y) \tag{2}$$

where $p_\theta$ and $q_\phi$ indicate that relevant components are approximated via neural networks. We derive the evidence lower bound (ELBO) for this model, starting with the data log likelihood:

$$
\begin{aligned}
log\, p(x|y) = log \int_z \sum_{c \in \mathcal{C}_y} p(x, z, c \mid y)\, dz &= log \int_z \sum_{c \in \mathcal{C}_y} q(z, c \mid x, y) \frac{p(x, z, c \mid y)}{q(z, c \mid x, y)}\, dz \\
&\geq \int_z \sum_{c \in \mathcal{C}_y} q(z, c \mid x, y)\, log \frac{p(x, z, c \mid y)}{q(z, c \mid x, y)}\, dz \quad \text{(Jensen's Inequality)} \\
&= \mathbb{E}_{q(z,c|x,y)}[log\, p(x, z, c \mid y) - log\, q(z, c \mid x, y)] \\
&= \mathbb{E}_{q(z,c|x,y)} \left[ \underbrace{\log \frac{p(c \mid y)}{q_\phi(c \mid x, y)}}_{\text{-KL}(c)} + \underbrace{\log \frac{p_\theta(z \mid c)}{q(z_\phi \mid x, c, y)}}_{\text{-KL}(z)} + \underbrace{\log\, p_\theta(x \mid z)}_{\text{Reconstruction}} \right] \\
&\coloneqq L_{\text{ELBO, L-GMVAE}}
\end{aligned}
\tag{3}
$$

**Model architecture and training.** In practice, the L-GMVAE consists of two neural network components, the inference model $q_\phi$ and the generative model $p_\theta$, matching the above definitions. The inference model takes $x$ and the known $y$ label as inputs, and outputs the probability distributions over the clusters. These are then concatenated together to predict $z$. The generative model produces a reconstructed input using $z$, and learns the Gaussian mixture prior ($p_\theta(z \mid c)$). The ELBO objective has three terms, Kullback–Leibler (KL) divergence for $c$ and $z$ as regularisers, and a reconstruction

term. The KL terms are expressed analytically, while the reconstruction term is the MSE loss between $x$ and the reconstructed $x'$. The practical loss function is a negated, weighted sum of these ELBO terms. See Appendices B and C for more details of our trained L-GMVAE models.

**Handling categorical data.** Additionally, L-GMVAE can accommodate one-hot-encoded (OHE) categorical data by simply adding a sigmoid function at the decoder's final layer dimensions which correspond to the categorical dimensions. During training, the reconstruction loss for these dimensions uses binary cross-entropy loss. During inference, a rounding operation is placed after the sigmoid output to determine an integer value of either 0 or 1 for each OHE dimension.

We next discuss three key desiderata of a well-optimised L-GMVAE which would make it an ideal tool as a recourse generator, motivating LAPACE. A recourse task is typically instantiated on a dataset $\mathcal{D} = \{x^i, y^{*,i}\}_{i=1,\ldots,N}$ and a trained classifier $M$. In our framework, L-GMVAE is trained and tested using the predicted dataset by the classifier, $\{x^i, M(x)^i\}_{i=1,\ldots,N}$, instead of the original dataset, in order for the explanations to be faithful to $M$.

**Meaningful centroids.** Each cluster centroid (the mean of its Gaussian prior) evolves into a representative point of its assigned class in the latent space. The synthetic point should also be representative of the class after being decoded into the input space. This is enforced by two components – the reconstruction loss ensures that the decoded centroid is a valid class instance, while the KL of $z$ loss positions the centroid at the geometric centre of all data points assigned to that cluster.

This characteristic motivates using these reconstructed centroids as part of the CE generation method, because they guarantee the validity of the generated CEs. Furthermore, these points should be well within the data manifold. Therefore, if a classifier is updated, these prototypes are more likely to retain their predicted class, promoting plausibility and robustness to model changes for CEs (Pawelczyk et al., 2020b). Finally, if we enforce that all CEs converge to the vicinity of the centroids of one class, it would also enhance the robustness of CEs to input changes. These properties are all quantitatively measured in our experiments in Section 4.

**Diverse representations.** The model learns varied prototypes for each class, implicitly encouraged by the two regularising terms. The KL of $c$ penalises deviation from the uniform prior over a class's assigned clusters, promoting the use of all available clusters. Concurrently, the KL of $z$ term incentivises separation, as each cluster learns a unique prior mean to best model the data routed to it. This encourages the reconstructed centroids to be distinctly different from each other. We would therefore have a diverse set of CEs (also quantitatively evaluated in experiments) if all prototypes were involved in CE generation.

**Smooth latent manifold.** As is standard in VAEs, proximity in the latent space corresponds to perceptual similarity in the input space. The KL of $z$ and the reconstruction loss are responsible for this property, respectively regularising an organised latent space and forcing the decoder to be a smooth, continuous function to accurately reconstruct inputs from their latent representations. Therefore, closeness in the latent space would translate to closeness in the input space, which is evaluated as the proximity between the CE and the original input.

## 3.2 LATENT PATHS AND COUNTERFACTUAL EXPLANATIONS

Next, we describe how L-GMVAE can be used to generate paths of counterfactual explanations leading to high-quality recourse. After learning, the L-GMVAE's latent cluster centroids are fixed and available through the optimised prior distribution $p_\theta(z \mid c)$ by taking the mean of the Gaussian component, denoted as $z_{c_i}$ for each $c_i \in \mathcal{C}$. We refer to the L-GMVAE's decoder reconstruction functionality as $Dec : \mathbb{R}^h \to \mathbb{R}^d$. Then, for a target counterfactual class $y'$ and its associated clusters $\mathcal{C}_{y'}$, $Dec(z_{c_j})$ is the reconstructed cluster centroid for each $c_j \in \mathcal{C}_{y'}$. From the discussions in Section 3.1, these $Dec(z_{c_j})$ points carry suitable properties for recourse purposes

---

**Algorithm 1** LAPACE

**Require:** $x$, $y$, $y'$, $z_{c_j}$ for $c_j \in \mathcal{C}_{y'}$, $Enc$, $Dec$, interpolation steps $T = \{0, \ldots, 1\}$.
1: **Init:** $P \leftarrow \{\}$, $z_x \leftarrow Enc(x, y)$
2: **for** $c_j \in \mathcal{C}_{y'}$ **do**
3:     $\text{path}_j \leftarrow []$
4:     **for** $\tau \in T$ **do**
5:         $z_{x \to c_j, \tau} \leftarrow (1 - \tau)z_x + \tau z_{c_j}$
6:         Add $Dec(z_{x \to c_j, \tau})$ to $\text{path}_j$
7:     Add $\text{path}_j$ to $P$
8: **return** $P$

---

because they are valid, plausible, robust, and diverse. Additionally, they do not risk exposing existing data points given their synthetic nature.

LAPACE is described in Algorithm 1. For input $x$ with predicted label $y = M(x)$ and a target class $y'$ for its CE, we first obtain its latent representation $z_x$ via the *Enc*oder of L-GMVAE. (Line 1). Then, latent paths are obtained from $z_x$ to each $z_{c_j}$, $c_j \in \mathcal{C}_{y'}$ via linear interpolation, denoted as $z_{x \to c_j, \tau} := (1 - \tau) z_x + \tau z_{c_j}$ with finite steps $\tau \in [0, 1]$. During the walk in the latent space, we obtain paths of CE points as $\{Dec(z_{x \to c_j, \tau})\}$, for each $\tau$ value and each $c_j \in \mathcal{C}_{y'}$ (Lines 5 and 6). Points along the paths can then be tested with $M$ to obtain their predicted labels. Once the L-GMVAE is trained, LAPACE is simple and effective, eliminating the need for gradient-based optimisation in the latent space and avoiding a complex hyperparameter-tuning process (as the step size is relatively trivial) for the CE generation process.

Figure 2: Example CE paths found by LAPACE on MNIST dataset, for an input image of class 5 and a target label of 7. This L-GMVAE has 3 Gaussian clusters per class. Each row is a separate CE path, going from the reconstructions of the original input (the second image from the left, $\tau = 0$) to each reconstructed cluster centroid (the last image, when $\tau = 1$).

Figure 3: Continued MNIST example for the second cluster (the middle row in Figure 2). A requirement that a dash should not appear in the resulting CE images of class 7 is enforced on every image. For an input image size of $28 \times 28$, pixel values greater than a threshold of 0.01 at the 13th to 18th rows and the 8th to 14th columns (where the dash appears) are penalised via a loss function.

For intuitive visualisation, we show a CE example with image data on MNIST dataset (LeCun, 1998) in Figure 2. Details of this trained L-GMVAE are in Appendix B. We observe that the L-GMVAE learns diverse writing styles of digit 7 as the reconstructed latent cluster centroids. The reconstructed paths are also plausible evolutions from digits 5 to 7. Later points in the path (e.g., roughly when $\tau \geq 0.7$) are more likely to be classified as the target class by a classifier. In Section 4, we quantitatively demonstrate LAPACE's competitive performance on tabular datasets.

### 3.3 ACCOMMODATING ACTIONABILITY CONSTRAINTS

LAPACE also supports actionability constraints defining whether and how portions of the input space (e.g. individual input features) could be modified. Formally, an actionability constraint has form $x_i \bowtie \alpha$, where $x_i$ is the $i$th feature of an input $x$, $\bowtie = \{=, <, >, \leq, \geq\}$ and $\alpha$ is a constant. To model the satisfaction of these constraints, we use a a differentiable function $g(x_i)$, where $g(x_i) \leq 0$ indicates that the constraint is satisfied. For example, for a feature $x_i$ that must not be greater than or equal to a value $\alpha$, the function is simply $g(x_i) = x_i - \alpha$. Constraints on multiple features can be intuitively summed together.

At each $\tau$ step during the latent space interpolation, we evaluate whether the constraints are satisfied via $g(Dec(z_{x \to c_j, \tau}))$. If not, we iteratively correct $z_{x \to c_j, \tau}$ via gradient descent until the constraints are satisfied or up to a maximum number of corrections $N_{correct}$: $z_{x \to c_j, \tau} \leftarrow z_{x \to c_j, \tau} - \eta \cdot \nabla_z g(D(z_{x \to c_j, \tau}))$, where $\eta$ is a small learning rate and $\nabla$ is the gradient.

In our MNIST example (Figure 2), the second cluster learns a dash for digit 7. If we decide that the dash is unwanted in our CE, we can enforce such a constraint, as illustrated by Figure 3. In the new paths, high pixel values at the dash positions are mitigated, including for the intermediate images.

## 4 EVALUATION

In this section, we quantitatively evaluate the utility of L-GMVAE, and benchmark the quality of LAPACE along the common desirable properties against state-of-the-art baselines.

## 4.1 EXPERIMENT SETUP

**Datasets and classifiers.** We run our experiments on datasets which are commonly used in prior work (Karimi et al., 2023) and are advocated in the CE benchmarking library, CARLA (Pawelczyk et al., 2021). They are *heloc* credit (FICO, 2018), *wine* quality (Cortez et al., 2009), *adult* income (Becker & Kohavi, 1996), and *compas* recidivism (Julia Angwin & Kirchner, 2016). We split each dataset into a train and a test set for training classifiers. To demonstrate the model-agnostic nature of our approach, we train a random forest and a neural network for each dataset, which we use to compute the CEs. Characteristics of the datasets and models are summarised in Table 1.

| Dataset | Classifier | Utility: Train on Real vs. Synthetic | Centroid Acc. |
|---|---|---|---|
| heloc | RF $73.05\%$ | $73.97\%_{\pm1.07\%}$ / $71.07\%_{\pm1.24\%}$ | $100\%$ |
| 23=23+0 | NN $73.58\%$ | $91.82\%_{\pm0.55\%}$ / $89.99\%_{\pm0.50\%}$ | $100\%$ |
| wine | RF $75.13\%$ | $89.70\%_{\pm1.66\%}$ / $87.42\%_{\pm1.46\%}$ | $100\%$ |
| 11=11+0 | NN $76.95\%$ | $85.88\%_{\pm0.85\%}$ / $84.21\%_{\pm1.68\%}$ | $100\%$ |
| adult | RF $81.15\%$ | $93.82\%_{\pm0.94\%}$ / $81.13\%_{\pm3.81\%}$ | $100\%$ |
| 13=6+7 | NN $82.04\%$ | $88.83\%_{\pm1.16\%}$ / $75.86\%_{\pm3.31\%}$ | $100\%$ |
| compas | RF $78.17\%$ | $90.79\%_{\pm1.63\%}$ / $85.03\%_{\pm1.54\%}$ | $100\%$ |
| 7=3+4 | NN $77.70\%$ | $92.26\%_{\pm1.70\%}$ / $84.00\%_{\pm3.26\%}$ | $100\%$ |

Table 1: Dataset, classifier details, and L-GMVAE utility evaluation. The first column reports the number of features (#continuous + #categorical) of each dataset. The test accuracies of classifiers are included. The Utility columns report the test accuracies of new classifiers following TSTR.

**L-GMVAE training and evaluation.** The train set is further split into a train and a test set for L-GMVAE training and evaluation. An L-GMVAE is obtained for each dataset-classifier pair, learning the classifier's predictions on the new training set. Each L-GMVAE has 5 clusters per class. See Appendix C for more details. Following the *Train on Synthetic, Test on Real (TSTR)* Protocol commonly employed to evaluate the utility of generative models Stoian et al. (2025), we sample a new synthetic dataset from the L-GMVAE matching the size of the new train set. Then, we train new random forest classifiers respectively using the new train set and the new synthetic train set, and evaluate their accuracy on the new test set. Well-optimised L-GMVAEs should result in a synthetic train set with a similar distribution to the original data, yielding a small gap between the test accuracies of the new classifiers, indicating good utility. Additionally, we evaluate whether the original classifier indeed classifies each reconstructed cluster centroid as its dedicated class. This is crucial for guaranteeing that the CEs found by LAPACE are valid. The results are shown in Table 1. L-GMVAE shows promising utility (up to 2.8% accuracy gap) on continuous-only datasets. When more categorical features are involved, larger gaps become apparent. Nonetheless, the centroid accuracies are consistently 100%. Also, as we will see by the plausibility evaluation in Section 4.2, the synthetic data from all trained L-GMVAEs remains realistic. These results indicate that our L-GMVAEs are suitable for recourse generation purposes.

**CE evaluation procedure and metrics[1].** For each dataset-classifier pair, we obtain a CE test set of 100 test points predicted with label 0. We then use each method to generate CEs and evaluate on the following metrics. This process is repeated 5 times on different test sets. The procedures are in line with the existing literature. *Validity*: the portion of test points receiving valid CEs. *Proximity*: the average L1 distance between a test point and its CE(s). Lower distances indicate that the recourse prescribed by the CE would be easier to achieve by the end user, therefore it is more preferred (Wachter et al., 2017). *Plausibility*: the average local outlier factor (lof) (Breunig et al., 2000) of each CE. Lof identifies outliers by measuring the local density deviation of a data point with respect to its neighbours, and lower values indicate better proximities to realistic data, therefore better plausibility. *Diversity*: if a CE algorithm finds multiple CEs per test input, diversity is calculated as the average pair-wise (L1) distance between these CEs. Higher distances indicate that the CEs are more diverse. *Robustness to Model Changes*: we obtain 20 retrained classifiers using different selections of the dataset as the training set, and test on average how many CEs are valid on these new classifiers. *Robustness to Input Changes* evaluates the stability of generated CEs

---

[1]The evaluation metrics and baseline method implementations are adapted from CARLA library (Pawelczyk et al., 2021) and RobustX library (Jiang et al., 2025). All experiments were performed on a linux machine with an Intel Xeon w5-2455X CPU with 128GB RAM, and 2x24GB NVIDIA GeForce RTX 4090 GPUs.

when small norm-bounded perturbations are applied to the test input. We perturb the test input 10 times with small noises, generate new CEs for each of the perturbations, and report the average distance between the new CEs and the original CE (or max-set-distance (Leofante & Potyka, 2024) if the method supports finding multiple CEs per input). Smaller distances would mean better stability - finding similar CEs for similar inputs.

Additionally, on the two continuous-only datasets, we manually specify some actionability constraints (10 and 18 for wine and heloc) in the style of previous work in tabular data generation (Stoian & Giunchiglia, 2025). Each constraint specifies either a range for a feature value or the relation between two features (one greater than another) for the resulting CE. We randomly select 5 constraints for each test input, which are then enforced on a path of CEs and check if there is at least one CE satisfying all constraints while staying valid.

| Method | Val. | Prox. | Plaus. | Div. | M Rob. | In. Rob. |
|---|---|---|---|---|---|---|
| NNCE (Brughmans et al., 2023) | ✓ | ✓ | ✓ | - | - | ✓* |
| FACE (Poyiadzi et al., 2020) | ✓ | ✓* | ✓ | - | - | - |
| RobXCE (Dutta et al., 2022) | ✓ | ✓* | ✓ | - | ✓ | ✓* |
| DiCE (Mothilal et al., 2020) | ✓ | ✓ | - | ✓ | - | - |
| DRCE (Leofante & Potyka, 2024) | ✓ | ✓ | ✓ | ✓ | - | ✓ |
| **LAPACE - Ours** | ✓ | ✓ | ✓ | ✓ | ✓* | ✓ |

Table 2: The CE desirable properties addressed by each CE method in our experiments. ✓(*) indicates the property is explicitly taken into account (implicitly encouraged) by the method.

**Baselines.** We include state-of-the-art baselines covering the desirable properties of CEs, summarised in Table 2. Because LAPACE generates a path of CE points, to enable direct comparison with the baselines, we take CE points at three different locations in the path. *LAPACE-First* refers to the first CE point which turns the prediction label, when moving from the original input to the reconstructed centroids. *LAPACE-Last* refers to the last points in each CE path – the reconstructed centroids. Then, *LAPACE-Middle* is generated by decoding the latent vector located at the midpoint of the linear path connecting the latent representations of the previous two points.

For the actionability evaluation, we compare our path CEs with two path-based variants of FACE (Poyiadzi et al., 2020), an algorithm designed to obtain in-distribution CEs. FACE-Interpolation simply interpolates between the input and the FACE CE with 20 steps. FACE-Neighbours greedily selects the nearest neighbours from the training dataset, forming a path of 20 points from the input to the FACE CE. For these two baselines, due to the lack of an explicit way to incorporate constraint requirements, we directly modify the feature values to satisfy them and observe whether any CEs stay valid. In contrast, LAPACE allows building the constraints into CE generation, as introduced in Section 3.3 (LAPACE-constrained). To allow for a direct comparison to the baselines, we also directly modify the feature values in unconstrained LAPACE generation (LAPACE-naive).

## 4.2 EVALUATION RESULTS

**Benchmarking results.** The benchmarking results are visualised in Figure 4. All included methods, apart from DiCE, always find valid CEs. In terms of proximity, NNCE and DRCE consistently perform the best, with DiCE showing competitive performances on continuous-only datasets. Our methods, especially LAPACE-First, show comparable performances. Notably, LAPACE achieves the best plausibility results across all datasets and models, effectively managing the known tradeoff between plausibility and proximity. In terms of diversity, our methods outperform DiCE and DRCE on datasets with mixed data types but perform worse on the two continuous-only datasets.

Our method demonstrates strong performance on two robustness metrics. LAPACE-Last achieves perfect (100%) robustness to model changes and is guaranteed to be perfectly robust to input changes, as it always generates the same cluster centroid as CE for any input. LAPACE-Last is also competitive when compared with the robust baselines. It matches the proximity of RobXCE (robust to model changes) while achieving better plausibility, diversity, and a more efficient runtime. Compared to DRCE (robust to input changes), LAPACE-Last has notably better robustness and plausibility. Among our method's variants, robustness to model changes steadily increases from First to Last. This is expected, as the CE moves from a point near the original input to a more central point within the desired class. This shift improves plausibility and encourages that newly trained classifiers will likely predict it correctly. Finally, LAPACE is one of the fastest methods (second

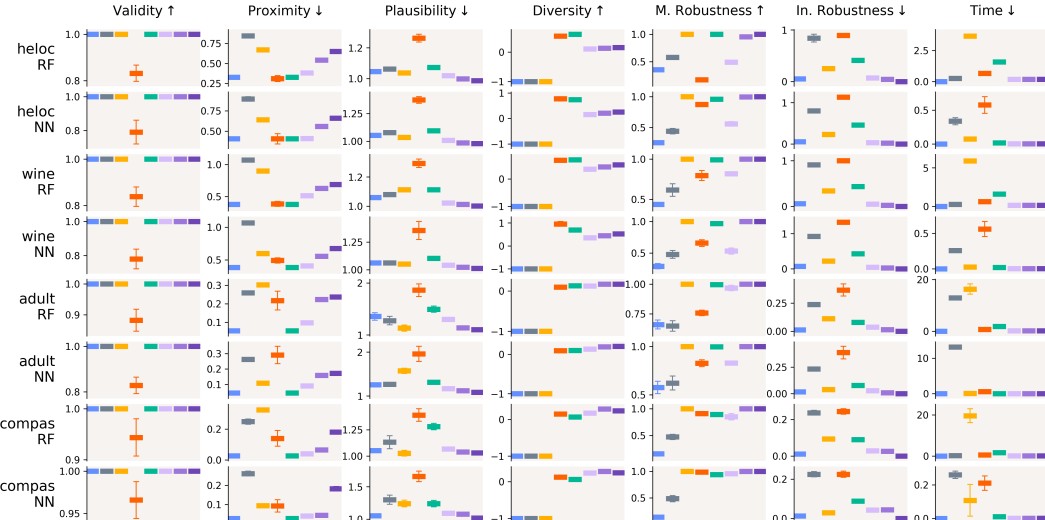

Figure 4: Quantitative evaluation of CEs generated by: **NNCE**, **FACE**, **RobXCE**, **DiCE**, **DRCE**, **LAPACE-First**, **LAPACE-Middle**, **LAPACE-Last**. Each subplot is the quantitative comparison of all methods on one dataset-classifier combination and on one evaluation metric. The arrows following each metric indicate that higher or lower values are considered better. For diversity, the first three methods find only one CE per input for which the evaluation metric cannot be computed. Therefore, they are assigned a value of -1.

only to NNCE) because of its amortised inference nature. After a one-time training cost for the main model (L-GMVAE), generating a counterfactual for any new input is very fast, requiring only a few forward passes. It also offers a key advantage over nearest-neighbour-based methods (NNCE, RobXCE, DRCE) by generating synthetic CEs, protecting the privacy of the original dataset.

| Dataset | FACE-Interpolate | FACE-Greedy | LAPACE-Naive | LAPACE-constrained |
|---|---|---|---|---|
| heloc-RF | 0.92 | 0.96 | 1.00 | 1.00 |
| heloc-NN | 0.82 | 1.00 | 1.00 | 1.00 |
| wine-RF | 0.74 | 0.88 | 1.00 | 1.00 |
| wine-NN | 0.90 | 1.00 | 1.00 | 1.00 |

Table 3: Actionability evaluations – portion of test inputs for which the CEs satisfy the actionability constraints while remaining valid.

**Actionability evaluation.** Table 3 reports the actionability results of LAPACE against two FACE variants. We used a combination of constraints on 5 features per dataset, which is around 20% and 50% of total features on heloc and wine, respectively. Under these conditions, both LAPACE variants consistently found valid CEs that satisfied all constraints, outperforming the baselines.

**More on the path points.** To further examine the characteristics of all generated path points connecting the input to LAPACE-Last centroids, we evaluate the classifier's predicted target class probabilities and the plausibility (LOF) of the paths. We present the results on heloc and wine in Figure 5. As $\tau$ increases, the class probabilities grow consistently, validating that the learned latent space of the L-GMVAE faithfully aligns with the classifier's predictions. Note that the probabilities at smaller $\tau$ values are highly dependent on the random selection of test inputs; therefore, variance could be high there. LOF scores for entire paths, including those before crossing the decision boundary, are consistently low (near 1.0), indicating that these synthetic points lie well within the data manifold. The reconstructed cluster centroids (LAPACE-Last points) often exhibit the greatest plausibility with LOF below 1.0, confirming they are located in regions denser than their neighbours.

In summary, LAPACE generates multiple paths of CEs that allow users to choose between solutions with high proximity and those with high plausibility and robustness. Actionability constraints can also be reliably addressed. The method is fast to compute, guarantees perfect robustness to input changes, and delivers the best-balanced performance across all metrics. Furthermore, the very low standard deviation across multiple runs confirms the stability of our approach.

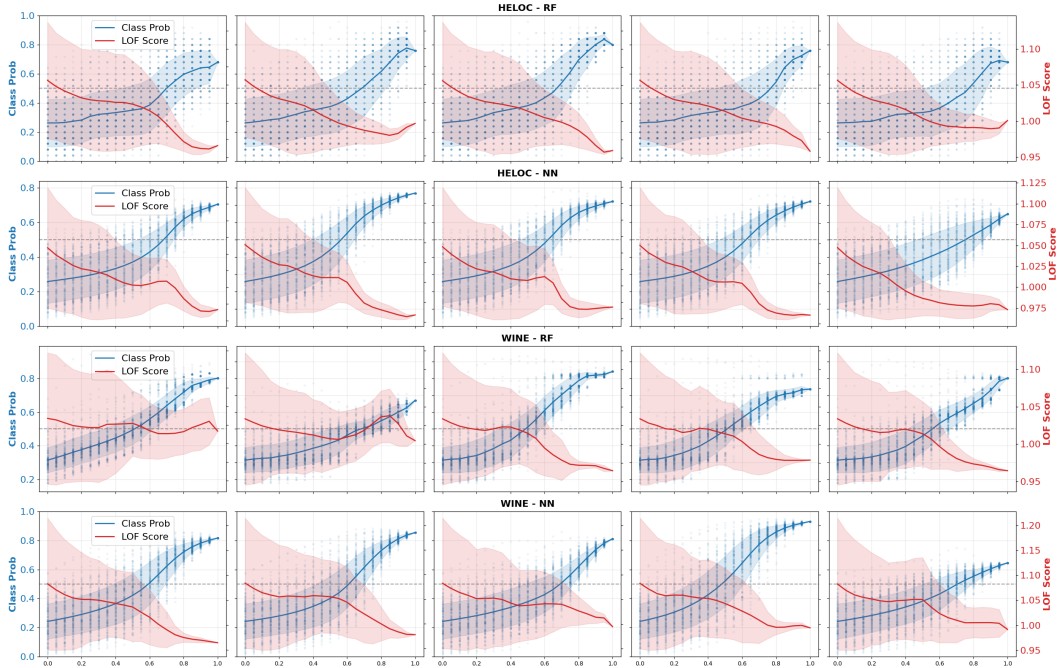

Figure 5: Classifier-predicted probability of the CE target class and plausibility evaluation for all path points with varying $\tau$. Each subfigure in a row shows the results for a path obtained from each L-GMVAE cluster for the desirable class. Points with a class probability greater than or equal to 0.5 (grey dotted line) are classified as the desirable class and could be used as CE points.

## 5 CONCLUSION

In this work, we introduce L-GMVAE, a generative model inherently suitable for recourse, and LAPACE, a novel, computationally efficient, and model-agnostic algorithm for generating CEs. LAPACE leverages the structured latent space of the L-GMVAE to produce diverse paths of CEs that are plausible, thus are robust to model changes. By design, all paths for a given target class converge to the same set of prototypical points, guaranteeing robustness against input perturbations. We also demonstrate that actionability constraints can be easily incorporated. Our comparative experiments using eight quantitative metrics validate LAPACE's competitiveness in generating high-quality CEs.

LAPACE has limitations worth discussing. First, the guaranteed validity of its CEs is contingent upon a successful L-GMVAE training. This requires a validation step beyond monitoring the loss, where one must confirm that the decoded cluster centroids are correctly classified by the original classifier. Second, in line with other VAE-based methods, our L-GMVAE is less effective on datasets with a large number of categorical features, as is reflected in utility results (Table 1). Nevertheless, the performance of our CEs on these heterogeneous datasets remains highly competitive.

This work opens several exciting avenues for future research. Establishing theoretical robustness guarantees on CEs obtained leveraging L-GMVAE would be valuable (Jiang et al., 2024b). Links to the synthetic tabular data generation literature (Stoian et al., 2025) are identified since it is how we view CE generation in this work. For example, dataset-level causal requirements between features (beyond user-specified actionability requirements) can be incorporated, possibly through adding specialised neural network layers into the training process of L-GMVAE (Stoian & Giunchiglia, 2025). Furthermore, the Gaussian mixture-regularised latent space learned by our model offers a powerful tool for extended forms of counterfactual explainability. Advanced interpolation methods such as Bures-Wasserstein interpolation (Bhatia et al., 2019) could be tested for replacing our linear one. Instead of generating point-based CEs for one input at a time, the structured manifold could be used to derive region-based explanations, providing insights into the model's behaviour for entire sub-groups of the data (Rawal & Lakkaraju, 2020; Bewley et al., 2024). Future work could also investigate extending L-GMVAE for synthesising CEs in other data modalities.

## ACKNOWLEDGEMENTS

Jiang, Rago and Toni were partially funded by J.P. Morgan and by the Royal Academy of Engineering under the Research Chairs and Senior Research Fellowships scheme. Leofante was supported by an Imperial College Research Fellowship. Rago and Toni were partially funded by the European Research Council (ERC) under the European Union's Horizon 2020 research and innovation programme (grant agreement No. 101020934). Any views or opinions expressed herein are solely those of the authors listed.

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

# A  Gaussian Mixture Variational Autoencoders

**GMVAE.** Given some input space $\mathcal{X}$, the GMVAE assumes that the observed variable $x \in \mathcal{X}$ is generated by a latent variable $z \in \mathbb{R}^h$ governed by a Gaussian mixture prior which has $K$ Gaussian components. A discrete variable $c \in \mathcal{C} = \{1, \ldots, K\}$ controls the cluster assignment for the mixture. The generative process of GMVAE is defined as follows:

$$p(x, c, z) = p(c)\, p(z \mid c)\, p(x \mid z) \tag{4a}$$

$$p(c) = \mathrm{Cat}(1/K) \tag{4b}$$

$$p(z \mid c) = \mathcal{N}(z \mid \mu_z(c), \sigma_z^2(c)) \tag{4c}$$

$$p(x \mid z) = \mathcal{N}(x \mid \mu_x(z), \sigma_x^2(z)) \tag{4d}$$

The inference model $q(z, c \mid x)$ of the GMVAE is factorised as:

$$q(z, c \mid x) = q(c \mid x)\, q(z \mid x, c) \tag{5}$$

where $q(c \mid x)$ is a categorical distribution and $q(z \mid x, c)$, the approximate posterior, is a Gaussian. Both are approximated using neural network models. Practically, the encoder takes in $x$ as an input, then predicts logits over the clusters, which are then concatenated with $x$ (or its processed version) to predict the Gaussian parameters $(\mu_z(c), \sigma_z^2(c))$ for sampling $z$. The standard reparameterisation trick is also applied at this step for differentiable training. Note that, unlike standard VAE, the prior distribution of $z$, $p(z \mid c)$, is also parameterised and has an associated learnable network component. This is updated via the KL divergence term (the second term in Equation 6 below). The evidence lower bound (ELBO) is:

$$
\begin{aligned}
L_{ELBO} &= \mathbb{E}_{q(z,c|x)}[log \frac{p(x, c, z)}{q(z, c \mid x)}] = \mathbb{E}_{q(z,c|x)}[log\, p(x, c, z) - log\, q(z, c \mid x)] \\
&= \mathbb{E}_{q(z,c|x)}[log \frac{p(c)}{q(c \mid x)} + log \frac{p(z \mid c)}{q(z \mid x, c)} + log\, p(x \mid z)]
\end{aligned}
\tag{6}
$$

# B  L-GMVAE in the MNIST Example

For illustration purposes, the L-GMVAE for the MNIST examples in Figures 2 and 3 is trained on the original MNIST dataset, instead of using any classifier's predictions. As mentioned in Section 3.1, practically the L-GMVAE loss function is a weighted combination of the two KL terms and the reconstruction term. For this particular model, the weights are 1:1:1 (unweighted). This model is trained with input size of 28x28 images, label dimension of 10, 3 latent clusters for each label for the latent Gaussian model, a learning rate of 1e-3 for the Adam optimiser and a batch size of 1024, with early stopping based on the validation loss.

Our network architecture is implemented as follows. The encoder network has two components. One takes in the input image and its one-hot-encoded label, produces a probability vector matching the cluster dimension through a 3-layer MLP component with ReLU activation and 512 hidden neurons, plus a softmax function. This is the $q(c \mid x, y)$ component in Equation 2 and is useful for predicting the assigned cluster for an input once the L-GMVAE is trained. Another component, $q(z \mid x, c, y)$ in Equation 2, takes in the input images, label, and cluster information, to produce the mean and logvar for the corresponding latent variable $z$, through a similar 3-layer MLP with two output heads. At inference time, this is useful for obtaining the latent encoding of an input.

The decoder network also has two components. One is responsible for the Gaussian mixture prior (Equation 1c), which simply takes in cluster information and outputs the mean and logvar of the latent variable via a linear layer. The second part implements the reconstruction functionality, mapping a latent vector back to the input space through a 3-layer MLP with ReLU activation and 512 hidden neurons.

More implementation details can be found in the accompanying code repository.

Figure 6 visualises the reconstructed cluster centroids for each digit class. We observe that this L-GMVAE model recognises different prototypical writing styles of each digit.



Figure 6: Reconstructed cluster centroids for the L-GMVAE model on MNIST dataset used in the examples.

## C   L-GMVAE IN THE CE EVALUATION

The architecture of the L-GMVAE models for Section 4 is identical to the one described in Appendix B, apart from different input and label sizes. The input sizes are decided by the dataset, and the label size matches the binary classification task. Each label is associated with five clusters. The training details are summarised in Table 4. The other details have been covered in the main text of the paper.

|  | Input Size | Latent Size | Batch Size | Learning Rate | Loss Weights |
|---|---|---|---|---|---|
| heloc-RF | 23 | 18 | 1024 | 1e-3 | 0.1, 0.1, 1 |
| heloc-NN | 23 | 15 | 1024 | 1e-3 | 0.1, 0.1, 1 |
| wine-RF | 11 | 8 | 64 | 1e-3 | 0.1, 0.05, 1 |
| wine-NN | 11 | 8 | 16 | 3e-4 | 0.5, 0.3, 1 |
| adult-RF | 13 | 10 | 1024 | 1e-3 | 0.4, 0.2, 1 |
| adult-NN | 13 | 10 | 1024 | 1e-3 | 0.4, 0.2, 1 |
| compas-RF | 7 | 5 | 512 | 5e-4 | 0.5, 0.3, 1 |
| compas-NN | 7 | 6 | 512 | 5e-4 | 0.5, 0.3, 1 |

Table 4: L-GMVAE training details. The loss weights are the weighting terms for KL($z$), KL($z$), and reconstruction.

## D   THE USE OF LARGE LANGUAGE MODELS

Large language model tools were used in a lightweight way to help structure the presentation of some paragraph drafts in the introduction and conclusion, at early stages of paper writing. They are not involved in further paper refinement. They are also used to aid in debugging the code implementations.

