# OpenReview forum: "Synthesising Counterfactual Explanations via Label-Conditional Gaussian Mixture Variational Autoencoders"
_ICLR.cc/2026/Conference — ICLR 2026 Poster_

### Official Review · Reviewer_7zY7 · 2025-10-25

**Soundness:** 3
**Presentation:** 4
**Contribution:** 3
**Rating:** 6
**Confidence:** 5

**Summary:**

A novel method to generate counterfactual explanations (CEs) is proposed. By learning a label-conditional VAE with Gaussian-mixture latent distribution (LGMVAE), the proposed method enables the generation of valid, plausible, close, robust, diverse and actionable CEs. The generation process only requires a few forward passes through the decoder of the LGMVAE (and one through its encoder) along an interpolation path, from the latent input to several possible cluster centroids, which correspond to the target class. Most costs are amortized through a single offline LGMVAE training. Extensive numerical experiments are provided.

**Strengths:**

- The proposed method is simple and effective.
- Most desirable properties of CEs are addressed.
- The method is clear and the paper is rather well-written.
- The numerical experiment section provides comparison against multiple baselines.

**Weaknesses:**

- Several questions regarding the interpolation process itself remain unexplored: I detail this in the “Questions” part of the review.
- Questions about the choice of target centroid are also unexplored. Indeed, although providing every interpolation path to every valid class centroid guarantees diverse explanations, one could wonder if smart target centroid choices can be made. For example, in Figure 1, choosing the centroid of cluster 4 would guarantee better proximity of the generated CEs than choosing that of cluster 3: such considerations could allow users to optimize for different criteria (in this example, a trade-off between diversity and proximity).
- Typo: L/K should be K/L at line 169.

**Questions:**

- How to better train a LGMVAE so that interpolation paths are realistic ? Regularization techniques to obtain a Euclidean latent space (where shortest paths are indeed straight lines) could be explored, e.g., regularizing the anisotropy of the decoder’s Jacobian.
- Differently, how to produce geodesic interpolations (in the sense of prefering traversing high-density regions) with a normally-trained LGMVAE ?
- A plot showing the evolution of the likelihood of the interpolated CE along the produced paths (as a function of \tau) is lacking. Since this is not optimized for, we can expect these interpolation paths to traverse low-density regions of the latent space (and thus of the feature space, once decoded).
- The predicted class (according to the model) along this path, as a function of \tau, would also be interesting, since the validity of these decoded interpolated points as CEs is not guaranteed.

---

> ### Author Response · Authors · 2025-11-27
>
> Thank you for the insightful feedback. We respond to the points raised in your review below:
>
> ### **Weakness 1 / Question 1, 3, 4**
> Thanks for bringing up this interesting point. We agree that it is important to ensure that the (reconstructed) interpolation paths are realistic points. We have not explicitly accommodated this consideration in the current work, but intuitively, learning meaningful Gaussian distributions should encourage the vicinity of each centroid to be realistic. The current hope is that the “borders” of the learned Gaussian clusters are close to each other, such that the interpolation paths could be realistic. Our original empirical evaluation results show that the LAPACE-first points (on the decision boundary) are comparably realistic to the baselines.
>
> **To make plausibility evaluations more comprehensive for other path points, we included the following new analysis, which also addresses your concerns in Questions 3 and 4. We added result plots in Figure 5 and relevant discussions at the end of Section 4.2.**
>
> We plotted the class probabilities, as well as the plausibility measure (LOF scores) for the desirable CE class (or "confidence", by the classifier) of all the path points against increasing $\tau$ values. We observe that the predicted probabilities steadily increase across the path as we move towards the desirable centroids. Also, the whole path has great LOF scores, with the centroids often showing even better plausibility. These new results highlight the usefulness of the learned L-GMVAE latent space and confirm our intuitions as stated above.
>
> ### **Question 2**
>
> Thanks for this very helpful feedback. It would be possible to perform advanced interpolation methods such as Bures-Wasserstein interpolation, which could better take into account the shapes of the Gaussian distributions. **We added a pointer to this in the future work section.**
>
> ### **Weakness 2**
>
> Generally, when diversity is considered, we are essentially offering options with trade-offs. The CE further away from the input could demonstrate other better characteristics, such as robustness, so not necessarily a bad thing. Functionalities on top of our method could be straightforwardly added to sanity-check the cluster centroids if we only want to use a subset of them.
>
> ### **Weakness 3**
>
> Thanks for pointing this out. **We have fixed them in the updated version of the paper.**

---

### Official Review · Reviewer_HV4K · 2025-10-31

**Soundness:** 3
**Presentation:** 3
**Contribution:** 3
**Rating:** 4
**Confidence:** 4

**Summary:**

This work presents a novel, model-agnostic approach for generating counterfactual explanations by constructing latent paths between factual instances and class-conditioned cluster centroids. These centroids are obtained using L-GMVAE, a label-conditional Gaussian Mixture VAE that learns multiple cluster centers per class. By interpolating in the latent space, the method allows users to control the trade-off between proximity and robustness when selecting counterfactuals. Robustness is particularly relevant in the current landscape of counterfactual explanations, where stability under input perturbations and model updates remains a key challenge. Additionally, the method supports actionability constraints by applying gradient-based adjustments whenever a generated sample violates a constraint.

**Strengths:**

1. The paper is generally well written, and the proposed method is clearly described.
2. The solution is model-agnostic and can be applied to both differentiable and non-differentiable models.
3. The approach allows users to choose an appropriate trade-off between proximity, plausibility, and robustness.
4. The authors introduce a novel L-GMVAE architecture that is specifically designed to support the proposed counterfactual generation method.

**Weaknesses:**

1. Evaluating plausibility using a single metric may be insufficient. Relying solely on Local Outlier Factor may not fully capture realism; complementary metrics (e.g., Isolation Forest, reconstruction likelihood, or classifier confidence margins) could provide a more comprehensive assessment.
2. The experimental evaluation is relatively limited. Although several baselines are included, the number and diversity of datasets is small.
3. Increasing the number of diverse counterfactuals requires retraining the L-GMVAE with a different number of clusters, which introduces additional computational overhead and slows down the overall counterfactual generation pipeline.
4. The effectiveness of the method heavily depends on successful L-GMVAE training. When latent clusters do not align clearly with class structure (e.g., due to class imbalance or categorical noise), the guarantees regarding validity and robustness may break down.
5. The visualizations in Figure 4 are dense and difficult to interpret. Presenting the results in table format will improve readability.
6. The evaluation uses only 100 test samples per fold pair, which may be insufficient for more complex datasets such as Adult.
7. The description of actionability constraints lacks precision. It would be useful to provide a clearer formal definition.
8. The method performs less effectively on datasets with many categorical features. This limits applicability in common tabular domains.
9. The work lacks an ablation study evaluating how the number of clusters per class affects diversity, robustness, and plausibility. Since this parameter controls core method behavior, such analysis would strengthen the empirical grounding.
10. The experimental section does not include comparisons to recent plausibility-oriented generative counterfactual methods such as PPCEF or C-CHVAE. Including such baselines would provide a more meaningful reference for plausibility performance.

**Questions:**

- How could the proposed method be extended to support global or group-level explanations, rather than generating counterfactuals for individual instances?
- How does the number of clusters per class influence proximity, plausibility, and diversity of the generated counterfactuals? Is there a principled way to select this parameter?
- What is the computational overhead introduced by enforcing actionability constraints? How does this affect the latency of counterfactual generation in practice?
- Since generative models often struggle with datasets containing many categorical features, did you consider techniques such as dequantization to improve latent space learning for categorical variables?
- The paper empirically demonstrates robustness, but does the method admit a formal robustness guarantee? If so, could you provide a more explicit formulation or theoretical justification?

---

> ### Author Response · Authors · 2025-11-28
>
> Thank you for the detailed comments. We respond to all points raised in the questions and weaknesses below:
>
> ### **Weaknesses 1 and 10**
>
> While we agree that having more statistics can be helpful, we stress that our experimental setup closely follows standard practice in the literature. LOF has been used as the only metric to evaluate plausibility extensively, see for example:
> - [DACE, Kanamori et al., IJCAI 2020]
> - [Dutta et al., ICML 2022] (cited in paper)
> - [Jiang et al., AIJ 2024c] (cited in paper)
> - [ElliCE: Efficient and Provably Robust Algorithmic Recourse via the Rashomon Sets, Turbal et al., NeurIPS 2025].
>
> **To make plausibility evaluations more comprehensive, we included new plausibility results for all path points found by our method in Figure 5, and added relevant discussions at the end of Section 4.2.**
>
> We plotted the class probabilities, as well as the LOF scores for the desirable CE class (or “confidence”, by the classifier) of all the path points against increasing $\tau$ values. We observe that the predicted probabilities steadily increase across the path as we move towards the desirable centroids. Also, the whole path has great LOF scores, with the centroids often showing even better plausibility. We also observe LOF scores of less than 1 sometimes, which, by LOF definition, means that the evaluated CEs are located in more densely populated areas than its neighbours in the dataset. These results further highlight the usefulness of L-GMVAE models in faithfully capturing the classifier's predictions and their ability to learn realistic data distributions.
>
> **(Weakness 10)** The additionally suggested baselines were not developed to be diverse, robust, or to find path CEs. For that reason, we believe the baselines we considered, together with a model-agnostic metric such as the LOF score, give a more comprehensive view.
>
> ### **Weakness 2**
>
> We note that we originally included 8 evaluation metrics on distinct properties of CEs: validity, proximity, plausibility, diversity, robustness to model changes, robustness to input changes, time, (in Figure 4), and actionability (in Table 3). We also have separate evaluations for the utility of the L-GMVAE models (Table 1), and the newly added plausibility analysis on all path points (new Figure 5). These are done for 8 dataset-classifier pairs. We believe this would make an extensive empirical evaluation.
>
> In terms of the number of datasets, we included four, which is also the amount included in the CE benchmarking library, CARLA [Pawelczyk et al., NeurIPS 2021]. This is also aligned with the experimental procedures of prior work, for example:
> - 2 datasets in [DACE, Kanamori et al., IJCAI 2020]
> - 2 datasets in [Dutta et al., ICML 2022] (cited in paper)
> - 3 datasets in [Robust Counterfactual Explanations for Neural Networks With Probabilistic Guarantees, Hamman et al., ICML 2023]
> - 4 datasets in [Jiang et al., AIJ 2024c] (cited in paper)
> - 5 datasets in [ElliCE: Efficient and Provably Robust Algorithmic Recourse via the Rashomon Sets, Turbal et al., NeurIPS 2025].
>
> ### **Question 2, Weaknesses 3 and 9**
>
> Thank you for the insightful comments about the number of clusters. It is true that to alter the number of clusters in L-GMVAE, one would need to obtain new models. However, it is also possible to train one model with a large number of clusters, then develop some sanity-check functionalities to choose subsets of centroids, thereby avoiding the retraining. With our already trained L-GMVAEs, we could use *up to* 5 clusters. Our new results in Figure 5 show that the plausibility evaluations and classifier confidences of each path (corresponding to each cluster) are quite similar. This could indicate that the performance results are only mildly influenced with varying number of clusters. We leave further investigations to future work.
>
> ### **Weakness 4**
>
> This is true and was mentioned in the limitations part of Section 5 in our submission. We have explicitly discussed that the effectiveness of LAPACE depends on the successful training of the L-GMVAE models, and that a check needs to be performed to make sure the reconstructed centroids are correctly classified. In the context of our experimental setup, successful training was not an issue.
>
> ### **Weakness 5**
>
> Our evaluations in Figure 4 involve 7 evaluation metrics for 8 dataset-model combinations, making this current figure the most space-efficient way of presenting them. This way of presenting results was inspired by prior work in the area, see e.g. [Bewley et al., Counterfactual Metarules for Local and Global Recourse, ICML 2024].
>
>
> ### **Weakness 6**
>
> We note that we repeat 5 times on different test sets for each dataset, so that would be 500 points per dataset, instead of 100. We believe this number is sufficient to draw meaningful conclusions.

---

> ### Author Response · Authors · 2025-11-28
> **continued**
>
> ### **Weakness 7**
>
> Thank you for the suggestion. We have updated the paper with more formal definitions for actionability constraints, making it clearer. **These are added as the first paragraph of Section 3.3.**
>
>
> ### **Question 1**
>
> Thanks for this comment. We have originally included this in the future work part of Section 5. This could require some advanced sampling strategies and optimisation problem formulations in order to compute entire "counterfactual regions".
>
> ### **Question 3**
>
> It would require a few gradient descent updates. One can set the learning rate and a maximum number of steps allowed to control the time. Because these L-GMVAE models are very small neural networks, the computation time (for updating a single point) could generally be ignored. This step is also quite modular - we could run them only on the path points which are predicted as the desirable class.
>
> ### **Question 4, Weakness 8**
>
> We are aware of the recent literature on improved representation learning for tabular data, such as Entity Embeddings [Guo and Berkhahn, 2016] and differentiable relaxation techniques like Gumbel-Softmax [Categorical Reparameterization with Gumbel-Softmax, Jang et al., 2016] for handling discrete variables in VAEs.
>
> We would like to clarify that the L-GMVAE framework is modular and compatible with these advanced encoding schemes. For example, the input layer of our inference network (encoder) could be replaced with learnable embedding layers for high-cardinality features, and the decoder could utilise categorical distributions or Gumbel-Softmax relaxations to better model discrete outputs without altering the core logic of our proposed latent space structuring or the LAPACE algorithm.
>
> We adopted standard OHE encoding to maintain consistency with baselines and to ensure that performance improvements are driven by our proposed generative methods rather than advanced feature engineering.
>
> ### **Question 5**
>
> We do not provide theoretical guarantees for robustness against model changes in this work. Our intuition is that the strong robustness performance comes from a combination of strong plausibility and the high predicted class probability near the reconstructed centroids. The link for the former has been studied by [Pawelczyk et al., 2020b] (mentioned in the submission in the first paragraph of Section 2). For the latter, many existing works explicitly leverage raised class probabilities for better robustness with theoretical results; this is discussed in the recent survey [Jiang et al., 2024b] ("Increasing class scores" paragraph, Section 4.2).  Investigating theoretical robustness results would be exciting future work following this work, and **we have added this to the discussions in Section 5**.
>
> For robustness against input changes, we do have some apparent guarantees which are ensured by the LAPACE algorithm design - we always have the reconstructed cluster centroids among the CE points we return for every input.

---

### Official Review · Reviewer_uE7n · 2025-10-31

**Soundness:** 1
**Presentation:** 3
**Contribution:** 1
**Rating:** 2
**Confidence:** 4

**Summary:**

This paper proposes to provide counterfactual explanation using deep generative models (VAE with gaussian mixtures to be exact), by interpolating on the latent space between different latent clusters corresponding to different conditions/perturbations.

**Strengths:**

The paper is overall very easy to follow. Aside from a very minor clarity issue which I described in the next section, the methodology as well as mathematical formulation are generally laid out very clearly. The figures also gave a good demonstration of the key idea. The derivation of ELBO and it's underlying factorization are also expressed clearly.

**Weaknesses:**

I found the approach proposed in this work not well-justified, heuristic, and very much lack robustness if the authors decide to frame it under the counterfactual scope. The key concern is the blatant violation of the consistency assumption in causal inference, which is the fundamental backbone of the definition of counterfactual in Pearl's 3 layers of causality. To put it simply, the counterfactual needs to maintain the same exact exogenous noise as the factual, i.e. minimal changes under perturbation, i.e. anything that is not causality affected by the perturbation should remain exactly the same in the counterfactual. And there is nothing in this work's methodology that gives that kind of guarantee or at least that kind of encouragement. This aspect (usually called "exogenous noise abduction") is very important in deep counterfactual modeling, and I suggest the authors consult some literatures in counterfactual modeling [1,2,3] and especially the recent literatures [4,5] that specifically tackle this consistency issue, and made it more and more clear that VAE is not well-suited for deep counterfactual modeling [4,5]. To put it more simply in the context of this work, VAE essentially gives you these different latent clusters of points under different $c$, but you don't know which two points in two different clusters correspond to the same exogenous features. And when you do interpolation without knowing the optimal transport path, features not causally related to $c$ also change. To use MNIST for example, if you simply do interpolation from a point in a digit cluster to the center of another digit cluster, not only is the digit changed, the exogenous features (style of writing, thickness of writing, intensity of writing, etc.) could all change, which results in not minimal changes, which results in the wrong counterfactual explanation.

What the authors are doing in this work is better described as interventional (layer 2 of causality), not counterfactual (layer 3 of causality). The metrics used in experiments are also predominantly interventional inference metrics. I suggest the authors consult latest literatures in counterfactual evaluation [6] for the more valuable evaluations of counterfactuals. As for dataset, I also suggest evaluation on Morpho-MNIST instead of MNIST, which is a true perturbation dataset where the counterfactual truths can be simulated and the preservation of exogenous features can be evaluated.

Minor:

1. Clarity of the main objective: I suggest the authors parameterize Eq. (3) to make it more clear which components are parameterized by neural nets and which ones are assumed to be known priors. I found the paragraph underneath it (L195-202) to be slightly vague and not sufficient. Canonically, I would assume the learnt encoder components to be $q_\phi(c | x, y)$ and $q_\phi(z | x, c, y)$ and learnt decoder components to be $p_\theta(x | z)$, but in this case $p_\theta(z | c)$ is also parameterized and it'd be good to make it clear.

[1] Pawlowski, Nick, Daniel Coelho de Castro, and Ben Glocker. "Deep structural causal models for tractable counterfactual inference." Advances in neural information processing systems 33 (2020): 857-869.

[2] Shen, Xinwei, et al. "Weakly supervised disentangled generative causal representation learning." Journal of Machine Learning Research 23.241 (2022): 1-55.

[3] Ribeiro, Fabio De Sousa, et al. "High Fidelity Image Counterfactuals with Probabilistic Causal Models." International Conference on Machine Learning. PMLR, 2023.

[4] Wu, Yulun, Louis McConnell, and Claudia Iriondo. "Counterfactual Generative Modeling with Variational Causal Inference." The Thirteenth International Conference on Learning Representations.

[5] Ribeiro, Fabio De Sousa, Ainkaran Santhirasekaram, and Ben Glocker. "Counterfactual Identifiability via Dynamic Optimal Transport." arXiv preprint arXiv:2510.08294 (2025).

[6] Monteiro, Miguel, et al. "Measuring axiomatic soundness of counterfactual image models." The Eleventh International Conference on Learning Representations.

**Questions:**

N/A

---

> ### Author Response · Authors · 2025-11-27
>
> Thank you for the provocative feedback. We note that the problem setting we target is positioned within the literature on counterfactual explanations (CEs) for algorithmic recourse, which is a different research field than causal inference and causally inspired CEs. The recourse setting fundamentally assumes a machine learning classifier for the dataset at hand, which makes predictions that could potentially affect the status of the data subjects, such as a loan application. The CEs in this case need to be faithful to the classifier’s predictions in order to explain the model decisions, i.e., what changes are required by the classifier. The resulting CEs should be presented to the end users who might be negatively affected by a model decision, hence the desiderata of the CE points (Section 2, lines 119-134). This is different from the literature on causal inference that instead aims at causally characterising the data generation process.
>
> Causality-agnostic CEs have been widely studied by cognitive scientists and have been proven to serve a complementary goal when compared to causal explanations. For instance, Byrne observed that "People tend to construct causal explanations that refer to strong (necessary and sufficient) causes that co-vary with the outcome [...] whereas they tend to create counterfactuals that refer to background enabling (necessary
> but not sufficient) conditions that could prevent the outcome" [Byrne, IJCAI 2019, Counterfactuals in Explainable Artificial Intelligence (XAI):
> Evidence from Human Reasoning].  Such causality-agnostic CEs are typically used to create alternative scenarios that may elicit causal thinking, without necessarily requiring causal knowledge at generation time. This is true even in the context of recourse. A user whose loan application has been rejected may still want to play with the input application and try to identify alternative scenarios that would have led to a positive outcome (i.e. loan accepted), even if these alternatives may violate causal constraints. By this light, causality-agnostic CEs can be seen as a special case of causality-aware CEs where we intervene on all features.
>
> This also explains the use of the word counterfactual, as opposed to intervention. The question being asked here is: according to the classifier, what changes in the input x would one need, in order to have received a different classifier outcome? Differently, in the causality literature, the questions asked for a counterfactual would be: according to the true causal relationships among all variables, what changes in the input x would one need, in order to have observed a different y label. So, while we agree that under the causality lens, such forms of CEs considered in this paper are interventional, this paper aligns with an established research area that does not explicitly consider causal models of the data.
>
> **We have added a discussion along these lines to the paper in the second paragraph of Section 2. Citations to the mentioned causality papers are also incorporated here.**
>
> For the minor comment, **we have modified the equations and added notes to make it clearer.** The prior $p_{\theta}(z\mid c)$ is also a parameterised GMM model, unlike in the case of usual VAEs where the prior distribution can be expressed analytically. Such a formulation inherits from the original GMVAE works: [Dilokthanakul et al., 2016] and [Shu, 2016] (as cited in the paper).

---

### Official Review · Reviewer_XwU6 · 2025-11-02

**Soundness:** 3
**Presentation:** 3
**Contribution:** 2
**Rating:** 4
**Confidence:** 4

**Summary:**

The paper addresses the problem of generating counterfactual explanations (CE) that remain robust to small changes in both model parameters and input data. It introduces the Label-Conditional Gaussian Mixture Variational Autoencoder (L-GMVAE), which represents each target class as a set of clusters following a gaussian mixture distribution. The centroids of these clusters act as robust prototypes for the target class. Building on this, the authors propose LAPACE (Latent Path Counterfactual Explanations), which generates CE by interpolating between an input’s latent representation and the centroids of the target class, yielding diverse and plausible counterfactual paths. The method is benchmarked against several common baselines and datasets, demonstrating improved robustness, plausibility, and diversity compared to existing approaches.

**Strengths:**

- The L-GMVAE formulation is a good contribution, offering a unified framework that simultaneously addresses key aspects of robustness, validity, and diversity in CE generation.

- The experiments on L-GMVAE training and evaluation are well designed, providing convincing evidence that the learned latent structure captures the underlying data distribution effectively. This acts as an important sanity check, showing that the generative model produces samples closely resembling the true data, thereby validating its suitability for CE generation.

- The empirical evaluation demonstrates that the proposed LAPACE method performs strongly across multiple datasets. Results indicate that it is competitive or outperforms existing baselines, particularly in robustness and plausibility, highlighting the effectiveness of the proposed approach.

**Weaknesses:**

- The qualitative MNIST experiments are rather synthetic and do not fully demonstrate the method’s generality. It would be great if the authors could include results on more challenging image or text datasets. Similarly, the authors could incorporate more high-dimensional tabular datasets as well.

- The proposed approach for accommodating actionability constraints may risk moving latent representations away from the learned data manifold, potentially generating invalid counterfactuals. Some quantitative or qualitative assessment of this would be nice.

- The argument that cluster centroids lie well within the data manifold and improve robustness to model updates feels somewhat ad hoc. While there is supporting empirical evidence, a stronger conceptual or theoretical justification would be better,

- The presentation of results in Figure 4 is overly dense and makes it difficult to interpret. Simplifying the figure, or reorganizing the metrics for better readability, would help communicate the results more effectively.

**Questions:**

- Line 170, there seems to be a typo, it should be uniformly assign $K/L$  cluster to each class.
- Equation (3) probably has a typo, the LHS should be $p(x|y)$ instead of $p(x)$.
- Line 249, "Additionally, they do not risk exposing existing data points given their synthetic nature." I don't follow this, please clarify.

---

> ### Author Response · Authors · 2025-11-28
>
> Thank you for the insightful feedback. We respond to your points raised:
>
> ### **Question 1 and 2**
>
> Thanks for pointing out these typos. **We have fixed them in the updated version of the paper.**
>
> ### **Question 3**
>
> This refers to the fact that, the Dec($z_{c_j}$) points are synthetic points; therefore, they do not directly expose data points in the training dataset as some existing methods do. For example, NNCE and RobXCE would return real points in the dataset as their final CEs.
>
> ### **Weakness 1**
>
> We included the MNIST dataset mainly to illustrate the high-level idea of our proposed method. This is not intended to be an evaluation dataset. The scope of our paper is on CEs for recourse purposes in tabular classification tasks. The various high-level aspects (plausibility, diversity, robustness, etc.) are most meaningful for, and are mostly studied in, tabular recourse settings. Therefore, while there are concurrent works on CEs for images and texts, they are not for recourse purposes and are considered different literature. Nonetheless, adapting L-GMVAE and LAPACE for CEs in other data modalities would be exciting future work, **and we added this to the conclusion section**.
>
> The tabular datasets we used for experiments are common in prior work in CE. Adult, compas, and heloc are included in the CE benchmarking library CARLA [Pawelczyk et al., NeurIPS 2021] as three of their four advocated datasets. We substituted the other one for another continuous-only dataset, wine. **We additionally highlighted this at the start of section 4.1.**
>
> ### **Weakness 2**
>
> In the original submission, we have presented a quantitative evaluation of actionability. Specifically, we evaluate the constraints satisfaction rate after posing actionability requirements on a combination of 5 features. Setup details are described at the end of Section 4.1, and the evaluation results are discussed in Section 4.2. It is indeed the case that the gradient optimisation for actionability could potentially render the CEs invalid, but we have not observed this in our experiments.
>
> ### **Weakness 3**
>
> The problem of strong plausibility (within data manifold) leading to robustness against model changes has been theoretically characterised by [Pawelczyk et al., 2020b] (mentioned in the submission in the first paragraph of Section 2), and our empirical results reflect this intuition. Deriving theoretical robustness guarantees would also be an interesting future work for our method. **We additionally added relevant citations to the second paragraph of the "meaningful centroids" part in Section 3.1 to make it clearer. We also added this point to the future work in Section 5.**
>
> We consider the capability of learning realistic data distributions as already established in the generative deep learning literature. **In Section 4.2, we have added new results on the plausibility evaluation of all path points obtained in LAPACE.** We show that our whole paths are located well within the data manifold.
>
> ### **Weakness 4**
>
> Our evaluations in Figure 4 involve 7 evaluation metrics for 8 dataset-model combinations, making this current figure the most space-efficient way of presenting them. This way of result presentation is in line with prior work: [Bewley et al., Counterfactual Metarules for Local and Global Recourse, ICML 2024]

---

### Author Response · Authors · 2025-11-27
**Common response to all reviewers**

We would like to thank the reviewers for all your insightful feedback, which would help us improve this work.

We have updated our paper to address the comments. We made the following notable modifications:

- We **added a new experiment and relevant discussions in Section 4.2** (Evaluation Results), showing that all the points in our reconstructed CE paths have great plausibility performance, and that the **classifier's predicted class probabilities** (newly added evaluation metrics) for the desirable CE class steadily increase as we walk closer to the reconstructed cluster centroids (comments by reviewers HV4K, 7zY7).

- We **fixed two typos** spotted by reviewers XwU6 and 7zY7, and **added subscripts in parts of L-GMVAE formulations** to indicate which components are approximated by neural networks (comments by reviewer uE7n).

- We **added a discussion paragraph in Section 2 (related works)** to highlight the distinctions between our targeted counterfactual explanations literature within explainable AI research and the causal inference literature (comments by reviewer uE7n).

- We **incorporated the following points into the last paragraph of Section 5 (Conclusions) as additional future work**: theoretical robustness guarantees on CEs obtained leveraging L-GMVAE (reviewers XwU6, HV4K), using L-GMVAE for CEs in other data modalities (reviewer XwU6), and advanced interpolation methods to replace linear interpolation (reviewer 7zY7).

- We **included a more formal definition for the actionability constraint** to make it clearer at the start of Section 3.3. (reviewer HV4K).

We will post more detailed individual responses to each reviewer soon. Thank you for your time!

---

### Author Response · Authors · 2025-12-03
**Author final remarks**

Dear Area Chair and Reviewers,

We thank the area chair for further reviewing this paper, and the reviewers for their insightful feedback. We are encouraged that three reviewers (XwU6, HV4K, 7zY7) consistently recognised the core strengths of our work:

- **Novelty and Contribution**: The introduction of the Label-Conditional Gaussian Mixture VAE (L-GMVAE) and the model-agnostic method, LAPACE, for generating robust (against two forms of perturbations), diverse, and plausible counterfactual explanation (CE) paths for algorithmic recourse purposes has been seen as a major contribution.

- **Soundness and Clarity**: The methodology is mathematically sound, well-formulated, and clearly presented.

- **Strong Empirical Results**: The extensive evaluation across multiple datasets, baselines, and eight evaluation metrics demonstrates that our method achieves state-of-the-art performance, particularly in terms of robustness and plausibility.

We have updated the paper with one page of new content to address the reviewers' constructive feedback, including:

- **New Plausibility Analysis**: We added an experiment (Section 4.2, new Figure 5) demonstrating that all points along our generated CE paths maintain high plausibility (great local outlier factor scores) and show steadily increasing classifier confidence. This enhances our claims about the goodness of our proposed L-GMVAE model and the CE generation, LAPACE. We hope these address the concerns about the quality of all the interpolated points (separate from the three path points we evaluated in the original submission) (HV4K, 7zY7)

- **Clarifications and Formalisation**: We added formal definitions for actionability constraints (Section 3.3) and clarified mathematical notations in the generative model (XwU6, uE7n).

- **Broader Context**: We expanded the related work section to clearly distinguish our specific focus on algorithmic recourse for predictive classifiers from the separate field of causal inference, addressing the perspective raised by Reviewer uE7n.

Regarding Reviewer uE7n's specific concern about causal consistency: While we appreciate the theoretical depth of causal inference literature, our work targets the established problem of algorithmic recourse for standard machine learning classifiers. As highlighted in our response and supported by cognitive science literature (e.g., [Byrne, IJCAI 2019, Counterfactuals in Explainable Artificial Intelligence (XAI): Evidence from Human Reasoning]), recourse explanations serve a different but complementary goal to causal explanations, which is to identify the minimal changes required to alter a specific model's decision. Our method excels in our focused CEs-for-recourse literature, as confirmed by the other three reviewers.

We believe the revisions have strengthened the paper and that it now offers a valuable contribution to the research field of XAI.

---

### Meta-Review · Area_Chair_WKhN · 2026-01-07

**Summary:**

The paper proposes a Label-Conditional Gaussian Mixture VAE and a method called LAPACE to generate robust and plausible counterfactual explanations by interpolating latent paths. Reviewers generally found the method novel and the empirical results strong, particularly regarding robustness and the utility of the generative framework. Their initial concerns focused on the depth of evaluation, the density of presentation, and the definition of actionability constraints. One reviewer strongly objected to the approach on fundamental grounds, arguing that using VAEs for counterfactuals violates causal consistency and that the work should be framed as interventional rather than counterfactual. The authors provided a comprehensive rebuttal, adding a new experiment to analyze the plausibility and classifier confidence of the generated paths, clarifying definitions, and explicitly distinguishing their work on algorithmic recourse from causal inference literature.

**Reviewer Concerns:**

- Plausibility of Interpolated Points: The reviewers were concerned that points along the latent path might not be plausible. The authors added a new analysis (Figure 5) showing that LOF scores remain good and classifier confidence steadily increases along the path.

- Actionability Constraints: The authors provided a formal definition of the actionability constraints in the revision and clarified that their gradient optimization did not yield invalid CEs in experiments.

- Clarification of Notation: Minor errors and parameterization clarifications were fixed in the revision.

- Causal Consistency: Reviewer uE7n argued that the method lacks "exogenous noise abduction" required for true counterfactuals. The authors argued that algorithmic recourse is a distinct objective from causal discovery/inference

**Reviewer Scores:**

currently 2446. With 2's main concern being clarified by the authors that they are focused on a different domain, we ignore that in the final decision. I would expect the final score to be 456, borderline

---

### Decision · Program_Chairs · 2026-01-26

Accept (Poster)